# Changes in the Populations of Two Lymnaeidae and Their Infection by *Fasciola hepatica* and/or *Calicophoron daubneyi* over the Past 30 Years in Central France

**DOI:** 10.3390/ani12243566

**Published:** 2022-12-16

**Authors:** Daniel Rondelaud, Philippe Vignoles, Gilles Dreyfuss

**Affiliations:** Laboratory of Parasitology, Faculty of Pharmacy, University of Limoges, 87025 Limoges, France

**Keywords:** *Calicophoron daubneyi*, *Fasciola hepatica*, *Galba truncatula*, habitat, *Omphiscola glabra*, population, prevalence of infection, vegetation, water

## Abstract

**Simple Summary:**

Two parasitic diseases affecting humans and ruminants have a particular life cycle, because freshwater molluscs intervene in their transmission by ensuring the development of parasite larval forms. As the climate is changing, researchers have begun to investigate the effects of global warming on these snails and the larval forms of parasites that they harbour. Several authors have already conducted analyses of these diseases, but no field studies have been carried out so far. Therefore, in the present study, snail counts were conducted on 39 farms with acidic soils between 1976 and 1997, in 2013–2014, and in 2020–2021. The results showed that the number of snail populations decreased over time and that many populations have fewer and fewer individuals. This decline has also been faster in recent years. The infection of snails by one of the parasites has decreased over time. Conversely, snails are increasingly infected with the other parasite. These changes are due to the generalized use of a drug used to treat one of the diseases in ruminants and probably also due to the heatwave episodes that occurred for several years. As these larval forms are again infecting animals after their departure from the snails, practitioners must take these changes into account when treating ruminants for these diseases.

**Abstract:**

Field investigations were carried out during three periods (from 1976 to 1997, in 2013–2014, and in 2020–2021) on 39 cattle-raising farms on acidic soils to track changes in the populations of two Lymnaeidae (*Galba truncatula* and *Omphiscola glabra*) and their infection with *Fasciola hepatica* and/or *Calicophoron daubneyi*. Compared to the survey between 1976 and 1997 on these farms, there was a significant decrease in the number of the two lymnaeid populations and the size of the *G. truncatula* populations in both 2013–2014 and 2020–2021. This decline was significantly faster in the last nine years than it was before 2013. The area of habitats colonized by *G. truncatula* showed no significant variation over the years, while that of habitats with *O. glabra* significantly decreased in the period covered by the three surveys. The prevalence of *F. hepatica* infection in snails significantly decreased over the years, while *C. daubneyi* infection increased over time in both lymnaeid species. These changes are due to the use of triclabendazole to treat fasciolosis in ruminants since the 1990s, and are probably a consequence of the successive heatwaves that have occurred since 2018 in the region.

## 1. Introduction

Among the parasitoses that freshwater snails can transmit, two helminthoses, i.e., distomatosis caused by *Fasciola hepatica* and paramphistomosis caused by *Calicophoron daubneyi*, have a wide distribution in temperate countries. Fasciolosis has been recognized for its endemicity in Western Europe from many years [1,2,3]. This infection has increased in significance over the last 20 years, and it is currently considered a public health problem for humans [4,5,6]. Paramphistomosis is traditionally considered to have a limited veterinary significance, at least when farm animals are kept in good nutritional and sanitary conditions [6]. In Western Europe, this parasitosis did not develop until the 1990s, although it has long existed in domestic animals [7,8,9,10]. In the United Kingdom and Ireland, the prevalence of paramphistomosis in cattle has increased dramatically in recent years and may exceed that of fasciolosis in some areas [11,12]. Similar observations in cattle herds were also reported for acidic soils in central France with a decrease in fasciolosis (from 25.2% to 12.6% between 1990 and 1999) and a corresponding increase in paramphistomosis (from 5.2% to 44.7%) [13].

These two parasitoses are known to infect the same definitive host (usually cattle), and sometimes they can be found within the same individual [7]. Moreover, they are transmitted by the same intermediate hosts. In central France, the most common host snail is *Galba truncatula*, but another lymnaeid, *Omphiscola glabra*, which lives in the same meadows as *G. truncatula*, can also be an intermediate host, generally when it is co-infected by miracidia of *F. hepatica* and those of *C. daubneyi* [14,15]. Samples of adult *G. truncatula* (>4 mm high) were taken in central France between 1989 and 2000 to determine the prevalence of snail infection by either of these two parasites. Of a total of 18,791 *G. truncatula* collected during this period, the average prevalence of *F. hepatica* infection was 5.1% (with annual variations ranging from 3.3% to 7.2%), while that of *C. daubneyi* was 3.3%, with an annual increase in this parameter since 1990, reaching 5.3% in 2000 [13].

Since the 2000s, a decline in the number of these two lymnaeid populations has been noted in central France. It was also observed in the number of overwintering snails in many populations [16,17]. In 2020–2021, this decline continued [18]. According to Vignoles et al. [18], this decrease in 2020–2021 could be partly due to the occurrence of successive heatwave episodes during the summer months of 2018 to 2020. These results raised two questions: what was the rate of the decrease in the number and size of lymnaeid populations over the past 30 years? and what were the consequences of this decline in snail infection by *F. hepatica* and/or *C. daubneyi*? To answer the first question, a retrospective study was carried out on 39 farms raising beef cattle on the acidic soils in central France; our team conducted three surveys between 1976 and 1997, in 2013–2014, and in 2020–2021. The second question was answered by studying the prevalence of these two parasitoses in snails collected on the same farms during the three survey periods.

## 2. Materials and Methods

### 2.1. Farms Studied

The 39 farms are located in the northwest or west of the Haute Vienne department (Figure 1). Their altitude ranges from 155 to 241 m. These farms raised cattle on permanent grasslands not treated with lime, and the farming technique has not changed over the past 30 years. The subsoil of these farms is granitic or gneissic (Table 1). The 178 permanent meadows within the perimeter of these farms are hygro-mesophilous, with a mesophilous zone extending over most of the area in each meadow. These grasslands are subjected to alternating grazing by ruminants and mowing during the summer. A surface drainage network, either supplied by one or more temporary sources or not, is present in each pasture. This network is only cleaned up every three, four, or five years, depending on its state of degradation. Water, which flows through drainage systems or road ditches, is generally present from early October to late June or early July, with a maximum from January to late April. The pH of this water varies from 5.8 to 7, while the concentration of dissolved calcium ions is usually less than 20 mg/L [19]. These meadows are subject to a continental climate, strongly attenuated by the wet winds coming from the Atlantic Ocean. The mean annual precipitation ranges from 850 to 1100 mm per year, while the mean annual temperature ranges from 10.5 °C to 11.5 °C for most farms [20].

### 2.2. Animals

On the 39 farms, the average number of beef cattle varied from 34 to 57 depending on the year (Table 1). These ruminants grazed outside all year and were driven to different meadows depending on the growth of grass [21]. These farms are locally considered “fluke farms” because of the frequency of fasciolosis outbreaks that affected cattle herds prior to the 2000s. According to local veterinarians who worked with these breeders, until 1990, prophylaxis against fasciolosis was performed by the administration of closantel, albendazole, and/or nitroxynil [22]. From 1994, these drugs were abandoned in favour of triclabendazole [22], and farmers have gradually used this product to treat their cattle against fasciolosis (Table 1). This last drug was first used annually. As cases of bovine fasciolosis have become rarer on these farms since 2008, most breeders have subsequently treated their animals only in cases of clinical fasciolosis or with the detection of parasite eggs in cattle stools. As a result, the periodicity of this treatment has changed over time, with an interval of up to two years between two successive triclabendazole intakes for each animal. Paramphistomosis only developed on these farms in 1994 and was not treated until 2008. The discovery of a few cattle with diarrhoea led thirteen breeders (out of thirty-nine) to deworm their cattle with oxyclonazide once or twice in the past ten years.

The two Lymnaeidae lived in the same meadows and often on the same surface drainage network. The habitats of *G. truncatula* are generally located at the upstream end of each drainage swale or each man ditch, while the *O. glabra* populations generally live on the course of the same swales (Figure 2) with a distance of 8 to 25 m between the habitats of both lymnaeids [23]. The two snails also occupied other sites in these grasslands, such as slope rushes around temporary or permanent sources [23]. Pictures of these two habitat types have already been published in two reviews that our team has written on these species of Lymnaeidae [14,15].

On the different pastures, the habitats of Lymnaeidae were not separated by a fence from the rest of each meadow, so that cattle had access to snail habitats during their grazing.

### 2.3. Protocol for Snail Investigations

The 39 farms were selected for this study because an outbreak of animal fasciolosis was diagnosed by local veterinarians in their cattle herds between 1976 and 1997. Three surveys were conducted on each farm. The first investigations were carried out during the above period on the overall area of each farm to identify populations of Lymnaeidae, measure the area of their habitats, and count adult snails belonging to the generation born during the previous autumn (overwintering snails) in each population. Pastures were then re-examined in 2013–2014 and 2020–2021, according to the same protocol. The first two surveys were conducted in late March or early April. In 2020–2021, temperature leniency enabled surveys to be conducted from the end of February to the end of March on the meadows of the 39 farms. This period was selected because the snail habitats were then waterlogged and only populated by adults of the overwintering generation. When a snail population was not found in 2013–2014 or 2020–2021 in a meadow, the breeder was questioned to find out the possible cause of this disappearance.

Adults higher than 4 mm for *G. truncatula* and 12 mm for *O. glabra* were counted by sight or using a colander (mesh size, 3 mm), depending on the height of the water layer. During the first 2 surveys, each count was performed by 2 people for 30 to 40 min in habitats located along a drainage network or a road ditch and by 1 person for 15 to 20 min in those located on pond edges and stream banks. In 2020 and 2021, each count was performed by 1 person for 15 to 20 min per habitat (when there was a population). The area of each habitat was then determined. Measuring areas occupied by *G. truncatula* or *O. glabra* is easy in the case of habitats located in drainage swales or along the banks of ponds and streams. When the shape of the habitat was irregular, the only solution was to draw a map and determine the area of this habitat according to its shape and dimensions.

On the 39 farms, snail counts and the determination of habitat areas were always conducted by a member of our team and students who already had at least two years of experience in collecting and identifying Lymnaeidae in the field. On each farm, the values recorded in the different sites occupied by each snail species were pooled without taking into account the type of habitat.

### 2.4. Protocol for Detecting Parasite Larval Forms in Snails

As herds of cattle graze in turn on the meadows of each farm, according to the growth of the grass, snail sampling was conducted in the different habitats of lymnaeids in order to obtain a sample of *G. truncatula* and another of *O. glabra* for each farm and for each survey period. Samples of 100 *G. truncatula* each (between 1976 and 1997, and in 2013–2014) or 50 snails each (2020–2021) were randomly collected within the perimeter of each farm, taking into account the number of snail habitats and the number of overwintering snails in each population. The same protocol was used in the case of *O. glabra*. Collected snails were over 4 mm high for *G. truncatula* and over 12 mm high for *O. glabra.* After being transported to the laboratory under isothermal conditions, the snails were dissected under a stereomicroscope to detect the presence of larval forms of *F. hepatica* and/or *C. daubneyi*. Rediae and cercariae belonging to either helminth were recognized based on our experience in identifying the larval forms of parasites in these two lymnaeid species [24,25].

### 2.5. Meteorological Data

The average monthly temperatures were provided by the Bellac weather station because it is the closest to the farms studied. The series of values considered concerned the temperatures recorded by the station from 1981 to 2010, in 2013–2014, and in 2020–2021 [26].

### 2.6. Statistics

Six parameters were studied. The first was the average monthly temperature during the three periods of investigation. Three other parameters were the number of populations for each snail species, the area of their habitats, and the density of overwintering snails in each population. The last two parameters were the prevalence of snail infection with *F. hepatica* or *C. daubneyi*. These parameters were noted, taking into account the farm investigated, the snail species, and the investigation period. Individual values noted for the habitat areas of each snail species during each survey were therefore reduced to an average and framed by a standard deviation for each farm. A similar protocol was used for the densities of overwintering snails per m^2^ of habitat.

As these six parameters were recorded on the thirty-nine farms during three successive surveys, the data were first analysed using the Shapiro–Wilk normality test [27]. As the distributions of these values were not normal, Friedman two-way analysis of variance was used to establish the levels of statistical significance.

The decline in the number of populations was calculated using the ratio of the number of populations in 2020–2021 to that recorded between 1976 and 1997. A similar protocol was used to determine the decline in the habitat area and the number of overwintering snails in the different populations of Lymnaeidae.

## 3. Results

Figure 3 shows the average monthly temperature for each investigation period. Temperatures were not significantly different from one investigation period to another. However, there was a more pronounced difference between temperatures in August and September (*p* = 0.08) and another between temperatures in June, September, and October (*p* = 0.10). These increases in temperature were therefore close to significance in summer and early autumn.

### 3.1. Snail Populations

Table 2 shows the characteristics of snail populations during the three surveys. Compared to the values recorded before 1998 on the 39 farms, the total number of snail populations in 2020–2021 significantly decreased, by 54.2% for *G. truncatula* (Chi^2^ = 78.0; *p* < 0.001) and by 34.5% for *O. glabra* (Chi^2^ = 78.0; *p* < 0.001). This decline was faster during the last nine years than in the period between the first two surveys: a decrease of 34.6% for *G. truncatula* and 22.1% for *O. glabra*, compared to 30% and 15.9%, respectively. The average area of habitats occupied by *G. truncatula* did not show any significant variation between the three periods of investigation (Chi^2^ = 0.80; *p* = 0.67). In contrast, the average area of the *O. glabra* habitat significantly decreased, by 33.4%, between the three surveys (Chi^2^ = 20.2; *p* < 0.001). The number of overwintering snails per population showed a significant decrease of 79% for *G. truncatula* (Chi^2^ = 23.0; *p* < 0.001) and 75.3% for *O. glabra* (Chi^2^ = 14.2; *p* < 0.001) between the three surveys. The decrease in the size of each *G. truncatula* population was faster between 2013–2014 and 2020–2021 (69.2%) than in the period between the first two surveys (32.0%). In contrast, the rates of decline during the above two periods were close for *O. glabra*: 61.2% and 63.5%, respectively.

### 3.2. Prevalence of Parasite Infection in Snails

On the 39 farms studied (Table 3), the prevalence of infection with *F. hepatica* significantly decreased, by 95.9%, for *G. truncatula* (Chi^2^ = 37.2; *p* < 0.001) between the first and third surveys. For *O. glabra*, the situation was quite different. Several immature rediae of *F. hepatica* were noted in 11 snails prior to 1998, while rediae and cercariae of *Fasciola* were noted during the other two surveys, with a significant decrease in prevalence between 2013–2014 and 2020–2021 (Chi^2^ = 14.7; *p* = 0.006).

No snail infected with *C. daubneyi* (Table 3) was observed in samples collected between 1976 and 1997, regardless of the lymnaeid species. Infected snails were later noted with significant increases in prevalence for *G. truncatula* (Chi^2^ = 55.5; *p* < 0.001) and *O. glabra* (Chi^2^ = 31.8; *p* < 0.001) between 2013–2014 and 2020–2021.

## 4. Discussion

Several factors may explain the changes in the number and size of lymnaeid populations on the 39 farms studied. Most of these changes are related to human activity on the grasslands. According to Dreyfuss et al. [16,17], the mechanical cleaning of the surface drainage system and/or the extension of underground drainage in many grasslands may be responsible for the disappearance of many snail populations and a reduction in their size since the 1970s. Another more recent factor is climate change, with the succession of heatwave episodes over recent years resulting in a decline in biodiversity, which is linked to the extinction of numerous species (see review by Johnson [28]). Several authors have already analysed the impact that climate change will have on the development of fasciolosis and host snails in several countries [29,30,31,32] because this parasitosis is highly dependent on temperature and precipitation [33]. According to projections by Cordellier et al. [34], the increase in evaporation, the decrease in oxygen concentrations due to increased water temperatures, and changes in precipitation patterns are likely to affect the survival and reproduction of Lymnaeidae, as well as other freshwater molluscs in northwestern Europe. These climatic disturbances will also lead to changes in the distribution of native freshwater species, with some migrating to colder locations, while others may have limited distribution. Non-native species may take over and expand their distribution areas [34].

Our observations regarding the declines in the number of both lymnaeid populations and in the size of the *G. truncatula* populations are consistent with data reported by our team on many farms on acidic and sedimentary soils in central France [16,17]. The new insight offered by this study concerns the rapidity of this decline over the last nine years. Similarly, habitats with *O. glabra* showed a significant reduction in their mean area over the same period, while sites occupied by *G. truncatula* showed no significant variation in their area. Several assumptions can be made to explain these results. In our opinion, the occurrence of heatwave episodes in 2018, 2019, and 2020 [35,36,37] are likely at the root of the changes observed in the populations of both species. These successive episodes of heat waves over several years may have led to the progressive extinction of some snail populations and the numerical decrease in the number of individuals in other populations due to the death of many snails belonging to the spring generation during habitat aestivation. Under these conditions, the survivors would have laid fewer eggs in the fall, and there consequently would have been fewer overwintering snails in successive years.

The reduction in the area of *O. glabra* habitats over the past nine years can also be explained by the above hypothesis, assuming a lower number of snails in their habitat and, consequently, a lower occupancy of the habitat area. On the other hand, the lack of significant variation in the area of *G. truncatula* habitats must be related to their particular location in grasslands on acidic soils. According to Moens [38] and Vareille-Morel et al. [23], many *G. truncatula* habitats are located at the upstream end of swales, with or without a temporary source in the case of surface drainage or around temporary springs on the hills surrounding these grasslands.

The changes in the infection rates of both Lymnaeidae are more complex to interpret. The first factor is triclabendazole, which local breeders have used progressively since the 1990s to control *F. hepatica* infection in ruminants. This drug is known to be highly effective against adult forms of the parasite and also in juveniles up to 1 week old [39,40]. The use of this product has resulted in a sharp decrease in cases of cattle fasciolosis over the years [41,42]. As farmers switched from broad-spectrum anthelmintics to triclabendazole to treat fasciolosis in their cattle, the number of ruminants infected by *C. daubneyi* showed a marked increase over the years [7,13] because the two parasites are often present in the same animals. This also affected the infection rates of *G. truncatula* and *O. glabra*. The prevalence of infection with *F. hepatica* gradually decreased in *G. truncatula* over time (from 4.7% to 3.3% between 1990 and 2000 [13]), whereas that with *C. daubneyi* gradually increased (from 0.8% to 5.3% between 1991 and 2000 in the same grasslands [13]). The second factor is more difficult to comment on, as the 19 *G. truncatula* and 12 *O. glabra* infected with *F. hepatica* were only collected from 5 and 2 farms, respectively, while the grasslands of the 39 farms were investigated in 2020–2021 (Table 3). The most valid hypothesis would posit that the low prevalence of these lymnaeids could be a direct consequence of the sharp decrease in cases of animal fasciolosis found in central French farms [42]. However, unfavourable conditions for the survival of lymnaeids and the embryogenesis of parasite eggs cannot be excluded, because the snail habitats on many farms remained dry for a long period during the autumn months in 2018, 2019, and 2020, meaning that individuals of the overwintering generation would have been unlikely to be infected.

Lower prevalence values in *O. glabra* than in *G. truncatula* have already been noted by Abrous et al. [43,44] in farms on acidic or sedimentary soils when the two lymnaeids were living in the same environment (a surface drainage swale on acidic soil, for example). These results are related to the susceptibility of each lymnaeid because the miracidia of both parasites are more attracted to *G. truncatula* than to the other species [45,46,47].

## 5. Conclusions

There has been a significant decrease in the number and size of lymnaeid populations in central France over the past 30 years, and this decline was significantly faster in the last 9 years than before 2013. The prevalence of *F. hepatica* infection in *G. truncatula* has also decreased over the years, while *C. daubneyi* infection has increased over time in both lymnaeid species. These changes are due to the use of triclabendazole to treat fasciolosis in ruminants since the 1990s and are likely also a consequence of the successive heatwaves that have occurred in the region since 2018.

## Figures and Tables

**Figure 1 animals-12-03566-f001:**
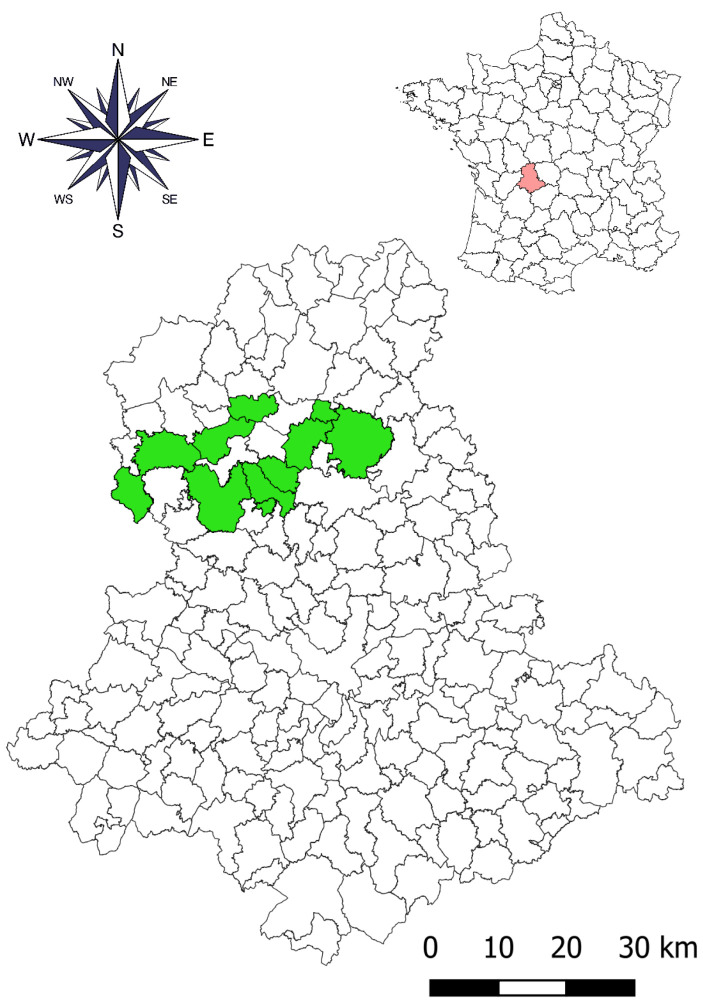
Geographical location of the Haute Vienne department in central France (upper map) and the municipalities (in green) on which the 39 farms are located (lower map).

**Figure 2 animals-12-03566-f002:**
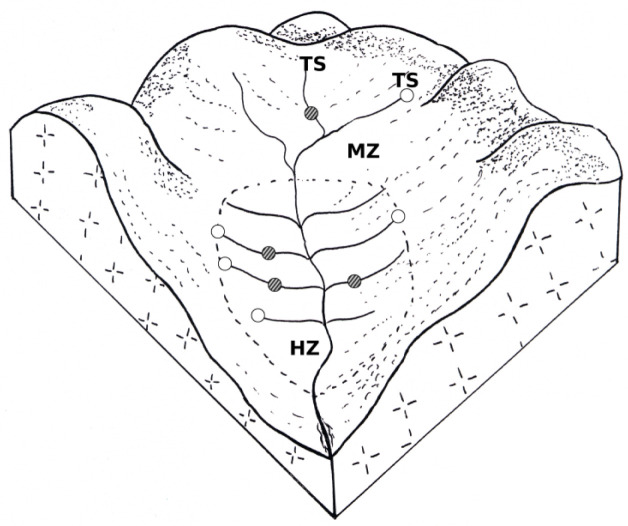
Block diagram showing the location of the most frequent habitats colonized by *Galba truncatula* (○) and *Omphiscola glabra* (●) in a central French grassland on acid soil. Most of these meadows have a surface drainage system to evacuate runoff. In the example shown in this figure, four swales on each side open into the main ditch. Two temporary sources (TS), located on the hillsides, discharge their water at the upstream end of the drainage ditch. HZ, hygrophilous zone; MZ, mesophilous zone.

**Figure 3 animals-12-03566-f003:**
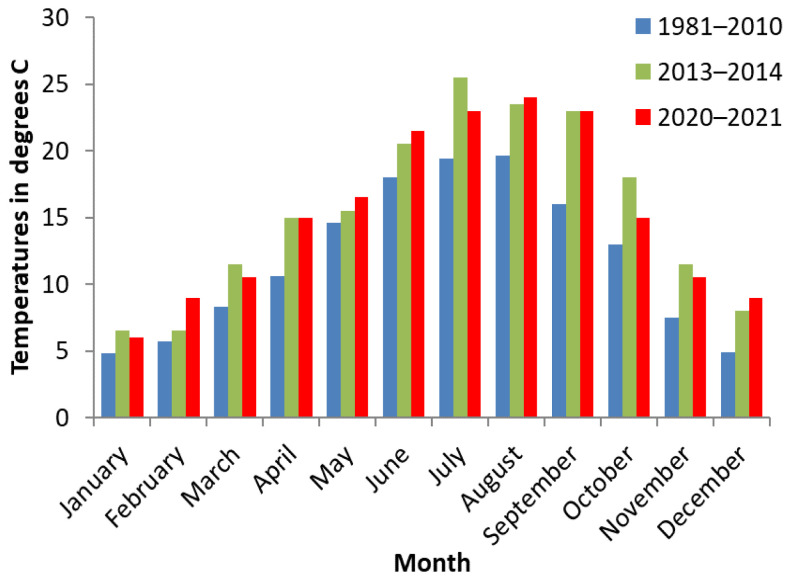
Average monthly temperatures recorded by the Bellac weather station from 1981 to 2010, in 2013–2014, and in 2020–2021.

**Table 1 animals-12-03566-t001:** Characteristics of the 39 farms selected for this study. All the farms raised beef cattle. Farms are numbered in this table according to the year of the first survey between 1976 and 1997. TBCZ, triclabendazole. * areas provided by the breeders; ** values noted during the first survey.

Farm No.	Soil Geology	Number of Cattle in 2020–2021	Year for the First Treatment with TBCZ	Total Area of Meadows (ha) *	Number of Snail Habitats **	
					*Galba* *truncatula*	*Omphiscola glabra*
1	granitic	41	1997	25	19	11
2	gneissic	49	1996	21	13	7
3	granitic	34	2001	28	19	9
4	granitic	51	1994	43	31	17
5	granitic	33	2003	21	16	11
6	gneissic	54	1994	31	14	9
7	granitic	57	1998	38	35	15
8	granitic	44	2002	34	22	13
9	gneissic	36	2003	19	17	9
10	granitic	48	1999	24	20	11
11	granitic	35	1994	35	16	10
12	granitic	41	1996	27	18	10
13	granitic	34	1995	29	20	14
14	granitic	52	1999	34	20	13
15	granitic	45	2000	27	11	8
16	gneissic	38	1995	20	15	11
17	granitic	49	2002	26	19	13
18	gneissic	44	2002	20	19	11
19	granitic	36	1996	31	17	13
20	granitic	42	1995	28	14	10
21	granitic	54	1994	50	33	21
22	gneissic	37	2000	23	19	12
23	gneissic	40	1998	34	15	8
24	granitic	47	2004	41	34	18
25	granitic	40	1994	33	16	10
26	granitic	38	1996	28	18	12
27	granitic	52	1995	38	26	13
28	gneissic	46	1996	35	28	15
29	granitic	40	2002	26	13	8
30	granitic	47	2000	49	39	17
31	granitic	35	1997	23	20	12
32	granitic	39	1995	42	32	15
33	gneissic	48	1997	34	14	10
34	granitic	42	2000	31	21	14
35	granitic	55	2001	41	27	13
36	gneissic	37	1997	24	17	13
37	granitic	42	1999	37	20	11
38	granitic	36	2002	31	15	10
39	gneissic	41	1998	42	27	12

**Table 2 animals-12-03566-t002:** Characteristics of snail populations on 39 farms on acidic soils across 3 surveys: between 1976 and 1997, in 2013–2014, and in 2020–2021. Mean values for habitat areas and the numbers of overwintering snails are given with their standard deviations.

Parameter	Period of Snail Investigations			Overall Rate of Decline (%)
	Before 1998	2013–2014	2020–2021	
Number of snail populations:				
*Galba truncatula*	809	566	370	54.2
*Omphiscola glabra*	457	384	299	34.5
Habitat area (m^2^):				
*G. truncatula*	1.8 (0.5)	1.6 (0.6)	1.7 (0.6)	0
*O. glabra*	8.1 (3.4)	8.9 (2.8)	5.4 (2.3)	33.4
Number of snails per population:				
*G. truncatula*	25.7 (9.4)	17.5 (7.3)	5.4 (1.6)	79.0
*O. glabra*	8.5 (3.2)	5.4 (1.6)	2.1 (1.2)	75.3

**Table 3 animals-12-03566-t003:** Number of adult snails infected with *Fasciola hepatica* or *Calicophoron daubneyi* and the prevalence of infection on 39 farms on acidic soils across 3 surveys. Snails collected were over 4 mm high for *Galba truncatula* and over 12 mm high for *Omphiscola glabra*.

Parameter	Number of Snails (Prevalence of Infection in %)		
	Before 1998	2013–2014	2020–2021
*Galba truncatula*			
Number of snails collected	3893	3897	1950
Number of snails with:			
*Fasciola hepatica*	198 (5.1)	82 (2.1)	19 (0.9) **
*Calicophoron daubneyi*	0	131 (3.4)	142 (7.2)
*Omphiscola glabra*			
Number of snails collected	3900	3900	1948
Number of snails with:			
*F. hepatica*	11 (0.28) *	55 (1.4)	12 (0.6) **
*C. daubneyi*	0	66 (1.6)	42 (2.1)

* Immature rediae only; ** On 5 farms (*G. truncatula*) and 2 farms (*O. glabra*).

## Data Availability

The datasets supporting the conclusions of this article are included in the article.

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
