# Peer review of "Changes in the Populations of Two Lymnaeidae and Their Infection by Fasciola hepatica and/or Calicophoron daubneyi over the Past 30 Years in Central France"

_animals, 2022, doi:10.3390/ani12243566_

Round 1
Reviewer 1 Report
This is a very interesting paper which highlights key trends for trematodes and their intermediate snail host in a specific region of France. In particular the authors hypothesise that climate change may negatively impact fluke, where the general consensus is that climate change actually has a positive effect on fluke (see Fox et al., 2011) Although there is great value in the information presented, I do not feel that this paper is publishable in its current state as certain key quantitative detail has not been included whilst the statistical analysis does not seem to be appropriate. If the authors can provide this extra information/data or clarify how descriptive information regarding farms were attained, correct statistical analysis and significantly improve grammar, this could be a highly valuable paper for this journal. Below I set out further detail on my concerns alongside suggestions for how the manuscript could be improved to become of sufficient standard for publication.
Methods – very detailed information is presented about each study farm. However, it is unclear how this information was gathered. Were farmers quantitatively surveyed or was the information gathered informally? In any case I believe this study would be greatly improved if a quantitative survey was conducted with each farmer. This would allow the authors to determine and quantify changes in management (i.e anthelmintics used, livestock and land management, and grazing strategies). Without this information I’m not sure if there is sufficient data to strongly back up the studies conclusions, for example it is unclear how the authors determined that farmers have changed anthelmintic groups used since the 90s or whether farms had or had not changed management practises over the course of the study period. At a minimum further details on how farm details were gained is needed.
Detailed description of habitats are also provided, however, I do believe the paper would be improved by the inclusion of photographs of the main habitat types, which would provide valuable context.
Methodology – it is unclear whether snails surveys were conducted by the same individuals or by different individuals. When reading the acknowledgements, it seems that different PhD students had conducted snail surveys in different years. This does raise some questions regarding the validity of the results, could differences in snails detection rates be influenced by the skill of the surveyor? I know from experience of supervising students, that certain individuals are highly skilled in finding Galba snails while others are less so. At a minimum this potential issue needs to be recognised in the methodology with clarification of whether these students were supervised in the field etc.
Statistics – more details are needed when describing the statistical tests as currently, unless further detail can prove me wrong, I believe they are not appropriate. For example, one assumption of the Kruskal wallis test is that each observation is independent. This is not the case here as the observations were repeated across three different periods (a repeated measures test would therefore be needed here). Similarly the Fishers exact test assumes independence within the observations, and a McNemar test would be needed to account for repeated measures. However, I am not convinced that contingency table based tests (i.e Fisher, McNemar) are appropriate as the data analysed by these tests (habitat surface area, snail density and parasite prevalence) would be continuous data. If I am wrong further detail is needed to explain the exact data used for each test. In all likelihood, complex multi level model would be needed to appropriately analyse this data. These models would be able to account for the repeated nature of the observations, the fact that some observations would have been nested within fields or farms, and covariables such as climate, sampling month, grazing history etc. that may have also influenced results.
It is interesting to see that the authors hypothesise that a series of recent heatwaves may have led to the recent decrease in snail numbers. I would like to see the inclusion of climate data to back this up and give the reader some perspective of how different these years were to average and historic values of rainfall/temperature etc. I.e were they major outliers or part of a long term trend. Would it be possible to calculate Ollerenshaw index values (or another fasciolosis risk model) to determine whether the changes observed on snail numbers concur with model predictions of fasciolosis risk? I also wonder if historic climate variables could also be used in this study to analyse associations between rainfall/temperature and snail density (it’s difficult to know if this is possible due to a lack of detail on when the first surveys were undertaken). But potentially a multivariate analysis could then be conducted which would identify if rainfall etc. are the primary associate of snail density and infection levels.
Furthermore, there are many minor errors in the text, listed below are some, however, the entire manuscript needs to be read carefully with the aim of improving grammar and removing errors.
Ensure all Latin species names are in italics
Line 47 – specify the role of grass/vegetation as well as water and vegetables
Line 48 – this sentence is incoherent and needs to be re written
Line 60-64 – this sentence could be improved to become more coherent.
Line 128 – error in the year ‘3008’
Author Response
Detailed list of responses. Manuscript 2035000.
General changes
All scientific names were italicized in the revised version.
The first paragraph of the introduction (present in the first version) was deleted in the revised version. The fourth paragraph was shortened and rewritten, indicating the questions at the origin of this study.
The first paragraph in the subsection 2.2 was partially rewritten and two additional references were added in the section References.
An additional subsection (2.5) on meteorological data was added in the revised version.
The subsection 2.6. was rewritten and the results of statistical tests were added in the section Results.
A table and two figures were added in the revised version.
Reviewer 1
The English text was checked by the following site: https://www.mdpi.com/authors/english (see certificate).
Methods: Information was collected from breeders who provided numerous documents (cattle health monitoring books, results of coproscopic examinations, etc) for this study. This information was confirmed by agricultural technicians who monitored pasture management on these farms and/or by local veterinarians who tracked cattle herds by delivering anthelminthics needed for parasite control.
Pictures of snail habitats: A sentence was added in the text of the revised version, indicating that pictures of this type have already been published in two monographs that our team has wtitten on these two lymnaeid species (lines 120-121 of the revised version).
Students: An additional sentence was added in the revised version (lines 148-150).
Statistics: This paragraph was rewritten taking into account another parameter (average monthly temperature during each survey) and the statistical results used to compare these data (lines 172-183). An additional paragraph on the origin of meteorological data was also added (lines 167-169).
Discussion: Discussion: Changes in average monthly temperature over the three surveys have already been incorporated into the revised version. The relationship between these results and other climatic parameters, grazing management, herd management, ... would require a further study and other surveys on the same farms to take into account future climate disturbances
Minor errors
Line 47 of the first version: The words 'vegetation' and 'water' were added in the list of key words.
Line 48, introduction: The paragraph 1 of the first version was deleted and the first words of this paragraph were placed in the paragraph 2 of the revised version (lines 47-49 of the revised version).
Lines 60-64, introduction: The text of these sentences was rewritten (lines 50-51).
Line 128: The correction was made in the revised version (line 108).
All changes in the revised version are highlighted in yellow.

Reviewer 2 Report
This research is being conducted to explore the population dynamics of two lymnaeid snails, Galba truncatula and Omphiscola glabra and their infection by liver flukes in central France. The population of lymnaeid snails has consequently declined since the last 30 years due to the treatment of fasiolosis in ruminants and heatwaves that have occurred since 2018. The findings of this study will have an impact on other fasciolosis endemic areas by assisting in the control of liver fluke transmission via snail hosts. The manuscript is well prepared and clearly describes the results and discussion.
Minor comment
- Scientific name should be italicized
- Please explain how to identify each lymnaeid snail species as well as parasites larval forms.
- Is there a hybrid Fasciola in this endemic area? If yes, how can the author define a species by observing radiae and cercariae stages?
Author Response
Detailed list of responses. Manuscript 2035000.
General changes
All scientific names were italicized in the revised version.
The first paragraph of the introduction (present in the first version) was deleted in the revised version. The fourth paragraph was shortened and rewritten, indicating the questions at the origin of this study.
The first paragraph in the subsection 2.2 was partially rewritten and two additional references were added in the section References.
An additional subsection (2.5) on meteorological data was added in the revised version.
The subsection 2.6. was rewritten and the results of statistical tests were added in the section Results.
A table and two figures were added in the revised version.
Reviewer 2
Identification of Lymnaeidae: The two species of Lymnaeidae were identified in the field because of the team's experience in studying these snails since the 1970s. The identification of these species was first made on the morphology of the shell and the study of the genital appearance as indicated by Germain (1930-1931) and other malacologists. This identification was subsequently confirmed using molecular biology based on the reports by the Mas-Coma team (Valencia, Spain) and the Hurtrez-Boussès team (Montpellier, France) on this subject.
Hybrids of Fasciola hepatica: No hybrid between F. hepatica and Fasciola gigantica has still been identified by molecular biology in central France.
All changes in the revised version are highlighted in yellow.
